# The Preparation, Thermal Properties, and Fire Property of a Phosphorus-Containing Flame-Retardant Styrene Copolymer

**DOI:** 10.3390/ma13010127

**Published:** 2019-12-27

**Authors:** Yu Sun, Yazhen Wang, Li Liu, Tianyuan Xiao

**Affiliations:** 1College of Chemistry and Chemical Engineering, Qiqihar University, Qiqihar 161006, China; benbenxiaoyu@hotmail.com (Y.S.); liuli520678@163.com (L.L.); 2Heilongjiang Province Key Laboratory of Polymeric Composition Material, Qiqihar 161006, China; 3College of Materials Science and Engineering, Qiqihar University, Qiqihar 161006, China; 4College of Light Industry and Textile, Qiqihar University, Qiqihar 161006, China; xtylwtg@163.com

**Keywords:** emulsion polymerization, flame retardancy, DOPO derivate, polystyrene, copolymer

## Abstract

A 9,10-dihydro-9-oxa-10-phosphaphenanthrene 10-oxide (DOPO) acrylate, (6-oxidodibenzo [c,e][1,2] oxaphosphinin-6-yl) methyl acrylate (DOPOAA), has been prepared. Copolymers of styrene (St) and DOPOAA were prepared by emulsion polymerization. The chemical structures of copolymers containing levels of DOPOAA were verified using Fourier transform infrared (FT-IR) spectroscopy and ^1^H nuclear magnetic resonance (^1^H-NMR) spectroscopy. The thermal properties and flame-retardant behaviors of DOPO-containing monomers and copolymers were observed using thermogravimetric analysis and micro calorimetry tests. From thermogravimetric analysis (TGA), it was found out that the T_5%_ for decomposition of the copolymer was lower than that of polystyrene (PS), but the residue at 700 °C was higher than that of PS. The results from micro calorimetry (MCC) tests indicated that the rate for the heat release of the copolymer combustion was lower than that for PS. The limiting oxygen index (LOI) for combustion of the copolymer rose with increasing levels of DOPOAA. These data indicate that copolymerization of the phosphorus-containing flame-retardant monomer, DOPOAA, into a PS segment can effectively improve the thermal stability and flame retardancy of the copolymer.

## 1. Introduction

Polystyrene (PS) is a very crucial thermoplastic and is widely used in many applications [1,2], such as automotive, housing, computer, packaging, and injection molding [3,4,5] due to its great mechanical properties and chemical stability. However, PS is flammable and it burns with the evolution of a large amount of smoke. Therefore, there has been a lot of effort to alter its unsatisfactory fire performance and promote its fire retardancy [6].

In the past, halogenated compounds were added into polymers as flame retardants. Although halogenated compounds are effective and commonly inexpensive, their use is coming under increasing regulatory pressure. At high temperatures, these materials were converted to volatile toxic dioxins [7]. Moreover, and more importantly, when items containing organic halogen flame retardants are discorded in a landfill the additives leach into the environment. As a result, human exposure to these additives and the resulting risk of disease are greatly increased [8,9]. Compounds containing nitrogen [10,11], phosphorus [12,13,14], and boron [15], which are more environmentally friendly as flame retardants, are being used. Phosphorus-containing flame retardants have received the most attention [16,17].

In 1972, DOPO was synthesized in a series of reactions. Since then, DOPO has been widely used in the field of polymer flame retardants. It may be easily modified, due to the P-H bond, to generate a series of derivatives that may be used to improve polymer thermal stability and fire behavior [18,19]. In addition, DOPO derivatives have been modified to contain different flame retardancy elements, such as P-N [20,21,22], P-Si [23,24,25], and P-N-S [26,27].

For PS, the frequently used means to ameliorate the flame retardancy is to add flame retardants. However, because of the poor compatibility with the polymer matrix and other disadvantages of DOPO and its small molecule derivatives, the application as a flame retardant for PS has been limited [28,29,30]. Compared with small molecule phosphorus-containing additive flame retardants, flame retardants that form an internal part structure show better stability and flame retardancy. Some phosphorus-containing flame-retardant monomers containing a C=C bond have been copolymerized with styrene to generate polymers with improved flame retardancy [31,32,33].

A DOPO acrylate, DOPOAA, has been synthesized as a flame-retardant monomer and copolymerized with styrene to generate a copolymer with reduced flammability compared that for PS. The thermal stability and flammability of the copolymer were improved by polymerization of DOPOAA into the structure of PS. A flame-retardant effect of DOPOAA in both the condensed phase and gas phase is suggested by results from TGA and micro-calorimetry.

## 2. Materials and Methods

The structure and properties of monomers and copolymers were established using ^1^H nuclear magnetic resonance (^1^H-NMR), Fourier transform infrared (FT-IR) spectroscopy, thermogravimetric analysis (TGA, New Castle, DE, USA), micro calorimetry (MCC, East Grinstead, West Sussex, UK) analysis, and the limiting oxygen index (LOI) test. ^1^H-NMR spectra were recording using a Bruker AV600 NMR spectrometer (Madison, WI, USA). 6-(hydroxymethyl)dibenzo[c,e][1,2]oxaphosphinine 6-oxide (ODOPM) was dissolved in dimethyl sulfoxide-*d*_6_, (6-oxidodibenzo [c,e][1,2] oxaphosphinin-6-yl) methyl acrylate (DOPOAA) was dissolved in deuterated chloroform, and the internal reference for ^1^H-NMR spectra was tetramethylsilane. FT-IR spectra were obtained using a Perkin Elmer Spectrum Two FT-IR spectrometer (Waltham, MA, USA) over the wavenumber range of 500 to 4000 cm^−1^. The sample and potassium bromide were mixed around and pressed to sheets for testing. Molecular weight M_n_ and M_w_ were measured using a Wyatt GPC/SEC-MALS gel permeation chromatography (Santa Barbara, CA, USA). The glass transition temperature (T_g_) was tested using a NETZSCH DSC (Selb, Germany) 204 F1 differential scanning calorimetry at a heating rate of 10 °C·min^−1^. Heat flow versus temperature scans from the second heating runs was plotted, and the glass transition temperatures (T_g_) were read at the mid-point of the inflexion curve resulting from the typical second heating. Thermal stability was determined using a TA Instruments Q5000 SA thermogravimetric analyzer, scans from room temperature to 700 °C at a heating rate of 10 °C/min under a nitrogen atmosphere. MCC was conducted using a FAA-PCFC micro calorimeter (East Grinstead, West Sussex, UK). Samples (5 mg) were heated up to 700 °C from the room temperature at the heating rate of 1 °C/s. The combustion furnace temperature averaged up to 900 °C and oxygen flow rate was 20 mL/min. LOI test values were obtained using a JF-3 oxygen index meter (Chengde, China) following the standard GB/T2406.2-2009. The size of the sample was 70 × 6.5 × 3.2 mm^3^.

The synthesis of ODOPM is shown in Scheme 1 [34,35]. 9,10-dihydro-9-oxa-10-phosphaphenanthrene 10-oxide (DOPO) (63.85 g, 0.3 mol) and 200 mL of toluene were added into the three-necked 500-mL glass flask equipped with a funnel, a condenser, a thermometer, and a magnetic stirrer, then heated to 80 °C under stirring. After that, 26.7 g of paraformaldehyde was fed in the flask in three batches over half an hour. The mixture was stirred for six hours at 95 °C [36]. The precipitate that formed was collected by filtration, washed several times with toluene, and dried to reduce pressure to constant weight. The yield of ODOPM was 91%.

The synthesis of DOPOAA is shown in Scheme 1 [37]. ODOPM (49.24 g, 0.2 mol), triethylamine (22.26 g, 0.22 mol), and 200 mL of methylene chloride were added into a three-necked 500-mL glass flask equipped with a thermometer, a magnetic stirrer, a condenser, and a funnel. The mixture was cooled and kept at 20 °C under stirring, then acrylyl chloride (19.91 g, 0.22 mol) dissolved in 50 mL of methylene chloride was added into the flask over 3.5 h. The mixture was first carried out for 2 h under 10 °C and then for another 6 h under room temperature. The precipitate was washed several times till the organic phase became neutral, and the crude product was obtained after the evaporation of dichloromethane. In the end, DOPOAA was purified from crude product by SiO_2_ column chromatography with ethyl acetate and petroleum ether as eluent, and the yield was 84%.

DOPOAA copolymer was synthesized as is shown in Scheme 1. Sodium dodecylbenzene sulfonate (0.75 g) and deionized water (80 mL) were added into a three-necked glass flask equipped with a thermometer, a magnetic stirrer, and a condenser, then stirred 30 min for emulsification at 70 °C. Thereafter, potassium persulfate (0.8 g), DOPOAA, and styrene were added to the nitrogen-protected flask. The reaction was carried out for 7 h and then kept in the air for 1 h before it was cooled to the ambient temperature. The saturated sodium chloride aqueous solution was eventually dropped into emulsion to adjust the pH so it became neutral and frozen for 24 h to obtain the flocculated precipitate. The flocculated precipitate was centrifuged and dried in the vacuum to obtain white powder.

## 3. Results and Discussion

### 3.1. Characterization of DOPOAA and DOPOAA-Styrene Copolymer

The structures of ODOPM, DOPOAA, and DOPO acrylate-styrene copolymer were confirmed by the help of FT-IR and ^1^H-NMR spectra. Figure 1 shows the FT-IR spectra of DOPO, ODOPM, and DOPOAA; the peaks at 1211 cm^−1^, 970 cm^−1^, 2894 cm^−1^, and 763 cm^−1^ belonged to P=O, P-Ph, C-H, and C=C-H respectively. Compared to the FT-IR spectra of DOPO, the intermediate, ODOPM, exhibited a brand new peak of -OH at 3218 cm^−1^. Additionally, the other groups’ peak positions coincided with the DOPO characteristic peaks. Thus, ODOPM was synthesized by the reaction through the change of functional group structure at the first step of the reaction.

In the spectrum of DOPOAA, the -OH peak at 3218 cm^−1^ appeared in ODOPM spectrum and disappeared due to the reaction occurred. Besides, the peaks belonging to C=C and C=O were shown at 1961 cm^−1^ and 1742 cm^−1^. It is indicated the hydroxyl group of ODOPM reacted with acryloyl chloride to form a new ester group.

Styrene and DOPOAA were copolymerized at different ratios to obtain a copolymer containing DOPO group, as shown in Table 1. In Figure 2, peak 3026 cm^−1^ belongs to the benzene ring, and 2924 cm^−1^ is the peak of -CH_2_. In Figure 2, compared PS-10 and PS-20 with PS, wherein the copolymer of C=O peak appears at 1734 cm^−1^, the peaks located in 1311 cm^−1^, 1181 cm^−1^ and 1028 cm^−1^ are attributed to P=O, C-O-C and P-O-C respectively. The peak of the flame-retardant group becomes more pronounced as the DOPOAA content increases.

Figure 3 is the ^1^H-NMR spectra of DOPOAA. In Figure 3, the peak at 7.30 belongs to CDCl_3_; the peaks at 7.22–8.09 are assigned to hydrogen atom in the special structure of the phenanthrene ring (H_a_); the peaks at 4.72–4.82 are assigned to the -CH_2_ of DOPOAA (H_b_); the peaks at 5.80–5.75 are assigned to the -CH (H_c_); and the peaks at 6.01, 6.03, and 5.65–5.69 are assigned to the -CH_2_ of the propylene, due to the chemical environmental asymmetry of the double bond. The above results confirmed DOPOAA has been successfully prepared by two-step organic modification of DOPO.

Figure 4 is the ^1^H-NMR spectra of the copolymer. In the copolymer, the peaks at 3.70 correspond to the -O-CH_2_-P- unit. Between 7.10 and 6.37, the aromatic protons appear. Signals of the main chain protons appear at 2.21–1.22 ppm. The compositions of the copolymers were calculated from the relative areas of the methylene and main chain protons resonance, using the following formula, where B_-CH_2_-_ and B_0_ were the relative resonance areas attributed to methylene and main chain protons. The DOPOAA content in the copolymer is shown in Table 1. It can be seen that as the DOPOAA content increases in the feed, the DOPOAA content in the copolymer also increases. At the same time, the content of DOPOAA in the copolymer is lower than that of the feed, which may be due to the large steric hindrance of the side chain. Data from gel chromatography showed that with the introduction of DOPOAA, the molecular weight of the copolymer increased.

The DSC curves of copolymer are shown in Figure 5. The results showed that when the content of DOPOAA in the copolymer increased, the T_g_ value of the copolymer decreased to 103 and 97 °C. This was because the introduction of DOPOAA made the copolymer obtain higher molecular flexibility [31,32], and DOPOAA had a significant effect on plasticizing the PS matrix.

### 3.2. Thermal Properties

TGA was used to investigate the degradation process and the thermal stability of DOPOAA and DOPO acrylate-styrene copolymer. The TGA curves of DOPOAA and its copolymers are displayed in Figure 6, and the test data, such as the temperature at 5% (T_5%_), the maximum degradation rate (R_max_), the temperature at the maximum degradation rate (T_max_), and the residue at 700 °C, are listed in Table 2.

Figure 6a,b are the TGA curves of DOPOAA and copolymers in nitrogen atmosphere from ambient temperature to 700 °C. Two decomposition stages of DOPOAA can been seen from the curves: the first stage is between 269 and 340 °C, and the second stage being between 347 °C and 451 °C. However, in the process of neat PS degradation, the sample began to lose weight at 365 °C and stopped decomposing at 450 °C, the R_max_ was −2.75%/°C when the temperature was 414 °C, and there was nearly no residue (0.21%).

At the same time, when the loading amount of DOPOAA increased, the residue at 700 °C of PS-10 and PS-20 raise to 0.89% and 0.91%. Compared with PS, the T_max_ of the copolymer hardly changed, but the R_max_ was reduced to −1.87 and −1.49%/°C, reducing 32% and 45.82%, respectively. The decrease of R_max_ indicates that a more compact char layer formed during the thermal degradation, and it could provide good insulation to the unburning part from the heat [21]. The change of residue at 700 °C and R_max_ indicated that the introduction of DOPOAA improved the thermal stability of the copolymer. In Figure 4, it can be found that the T_5%_ of PS-10 and PS-20 are lower than that of PS. This may be because the DOPO groups in copolymer catalyzed the polymer to dehydrate, and leads to more weight loss [38,39]. However, the residue at 700 °C of copolymer is higher, which indicated that copolymer has better thermal stability. Therefore, it can be considered that DOPOAA can effectively improve the thermal stability of copolymer.

### 3.3. Flame-Retardant Behaviors

MCC measurement and LOI test were used to evaluate flammability. The heat release rate (HRR) curves of PS and DOPO acrylate-styrene copolymer are shown in Figure 7, and the detailed data of MCC and LOI are listed in Table 3.

Figure 6 shows the HRR curves of PS and DOPO acrylate-styrene copolymer. As is shown in Table 3, the PHRR value of DOPO acrylate-styrene copolymer was much lower than that of the pure PS. Meanwhile, the higher DOPOAA content was, the lower PHRR value was. The PHRR value of the neat PS was 972 W·g^−1^, and the PHRR value of PS-20 was 354 W·g^−1^, which indicates that the addition of DOPOAA was conducive to improve the fire behavior of the copolymer samples. In LOI tests, the LOI value of the neat PS was only 18.1%, but the value of PS-20 rose to 24.3% at last, and DOPOAA content increased.

The phosphorus-containing groups in copolymer were thermally decomposed to form PO, phosphorus-containing radical [34,40]. The PO radical entering the gas phase can eliminate the radicals generated in combustion, thus inhibiting the further decomposition of the copolymer. These MCC and LOI data clearly demonstrate that the copolymerization of DOPOAA and styrene could improve the flame resistance of the PS.

## 4. Conclusions

A phosphorous-containing flame-retardant monomer, DOPOAA, was synthesized successfully. Then, its structure was characterized by FT-IR and ^1^H-NMR, and a series of the copolymers of St and DOPOAA were prepared at different ratios;TGA data showed that the residues at 700 °C rose with the increase of DOPOAA content, compared with the pure PS. In the MCC test, with the increase of DOPOAA, the PHRR value of copolymers was evidently reduced compared to pure PS, and the LOI value of copolymers increased. Judging, from the results of TG, MCC, and LOI tests, it is possible that the addition of DOPOAA could ameliorate the thermal stability and flame-retardant properties of PS. In conclusion, it is a tangible method to incorporate the flame retardancy monomer into the PS chain to prepare flame-retardant PS.

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
