# Peer review of "The Preparation, Thermal Properties, and Fire Property of a Phosphorus-Containing Flame-Retardant Styrene Copolymer"

_materials, 2019, doi:10.3390/ma13010127_

Round 1
Reviewer 1 Report
This paper describes the synthesis of a side chain containing phosphorus in a polystyrene copolymer, analysis by IR and NMR, and research that showed flame retardancy by thermal analysis.
Since no description of the problem was found for synthesis, identification, and thermal measurement, it should basically be accepted as it is.
The author may refuse because the purpose of this study was to show flame retardancy with new phosphorus side chain copolymers. However, compared with other known flame retardants containing phosphorus, it would have been good to discuss the superiority of this flame retardant effect.
That's all.
Author Response
Reply to Reviewer 1
By Yazhen Wang
Thank you very much for reviewing our paper. We carefully read your comments, which are quite useful. We have made suitable revisions and explain our revisions below. We hope this revision meets the criteria to be published in Materials. Thank you again for your constructive and helpful review.
Comment 1: The author may refuse because the purpose of this study was to show flame retardancy with new phosphorus side chain copolymers. However, compared with other known flame retardants containing phosphorus, it would have been good to discuss the superiority of this flame retardant effect.
Answer 1: Yes, we accepted your comments. The traditional additive flame retardant is easy to separated out and has poor compatibility with the substrate. Since the flame retardant synthesized by our group is introduced into polystyrene through copolymerization, the flame retardant can coexist with PS, the disadvantages of additive flame retardant are avoided. We have been carrying out comparative tests and adjusting the composition of the copolymer, but this process will take too much time, and the test results will be reported when we get all the data in the future.
Thank you again for your constructive and helpful advice.

Reviewer 2 Report
The submitted manuscript has potential to become interesting to a wider audience. Unfortunately, there are many shortcomings that need to be addressed. Firstly, the language and grammar is not at expected level. Authors used FT-IR and H-NMR methods to confirm the result of the synthesis. On the other hand, thermal properties of the investigated samples were determined utilizing TGA. However, the Fig. 4 is not in agreement with the conclusions made by authors (lines: 147-157). Fig. 4 clearly shows that samples 10%FR-PS and 20%FR-PS are thermally less stabile if compared to neat PS. Likewise, final mass of all samples seems to be practically equal. The Table 2, with probably useful data, is missing in the text. Therefore, based on the results showed in this work, authors cannot claim that “As a result, compared with the pure PS, the thermal stability and residue formation of copolymers containing DOPOAA were improved.”. TG measurements were done from room temperature to 700 °C at the heating rate of 20 °C /min under nitrogen atmosphere. It could be useful if authors would carry out these measurements at slower heating rate (10 °Cmin-1) under oxygen or air atmosphere. To evaluate the thermal stability utilizing TG, authors of this work can use many different criteria as follows: the onset temperature (Tonset), the temperature at 5% mass loss (T5%), the temperature at the maximum degradation rate (Tmax), the maximum degradation rate (Rmax), the final mass (mf) and the mass loss (Δm) for the corresponding degradation steps. “Unique maximum mass loss rate peak” and T5% alone are not enough. This section must be entirely modified.
Furthermore, authors evaluated flame retardant behavior (in the text it is written as “…Behaviors”) by MCC and LOI test. Indeed, the results of abovementioned test are in agreement and “These MCC and LOI data clearly demonstrate that the copolymerization of DOPOAA and styrene could obviously improve the flame resistance of the PS.”. Authors are advised to carry out the vertical flame test according to DIN 53906 standard method (if applicable) and to gain more data about flame resistance.
Author Response
Reply to Reviewer 2
By Yazhen Wang
Thank you very much for reviewing our paper. We carefully read your comments, which are quite useful. We have made suitable revisions and explain our revisions below. We hope this revision meets the criteria to be published in Materials. Thank you again for your constructive and helpful review.
Comment 1: The Table 2, with probably useful data, is missing in the text.
Answer 1: I’m so sorry, this is my negligence. Table 2 has been added into the revised paper (line 168), which contains detailed data of thermogravimetric analysis, including the temperature at 5% mass loss (T5%), the maximum degradation rate (Rmax), the temperature at the maximum degradation rate (Tmax) and the amount of residue at 700°C.
Comment 2: Fig. 4 clearly shows that samples 10%FR-PS and 20%FR-PS are thermally less stabile if compared to neat PS. Likewise, final mass of all samples seems to be practically equal.
Answer 2: From the Table 2 added in revised paper (line 168), the residue at 700°C of copolymer increased to 0.89% (PS-10) and 0.91% (PS-20), respectively, higher than the value of neat PS (0.21%).
As the literatures (No.38 and No.39 in reference list) reported, the polymer was catalyzed to dehydrate by DOPO-containing groups, which led to lower T5% and more mass loss in a certain range, but more residual at 700°C was obtained through adding more DOPO-containing group. The stability of the copolymer was improved by the addition of DOPOAA.
Comment 3: TG measurements were done from room temperature to 700 °C at the heating rate of 20 °C /min under nitrogen atmosphere. It could be useful if authors would carry out these measurements at slower heating rate (10 °C min-1) under oxygen or air atmosphere.
Answer 3: I'm sorry, our thermogravimetry test was actually performed at the heating rate of 10 °C /min. Due to clerical errors and negligence in proofreading, it was written as 20 °C /min. In the revised paper, it has been modified as the heating rate used in the actual test. Thank you again for giving us the opportunity to correct our negligence that ignored in the manuscript.
Comment 4: To evaluate the thermal stability utilizing TG, authors of this work can use many different criteria as follows: the onset temperature (Tonset), the temperature at 5% mass loss (T5%), the temperature at the maximum degradation rate (Tmax), the maximum degradation rate (Rmax), the final mass (mf) and the mass loss (Δm) for the corresponding degradation steps. “Unique maximum mass loss rate peak” and T5% alone are not enough. This section must be entirely modified.
Answer 4: Yes, you are right. In the Table 2, we listed the data of T5%, Rmax, Tmax, and residue at 700 °C. These data also indicate that the introduction of DOPOAA could improve the thermal stability of copolymer.
We have rewritten the section of Thermal Properties follow your comments (line 156-183) like this:
3.2 Thermal Properties
TGA was be used to investigate the degradation process and the thermal stability of DOPOAA and DOPO acrylate-styrene copolymer. The TGA curves of DOPOAA and its copolymers are displayed in figure 5, and the test data, such as the temperature at 5% (T5%), the maximum degradation rate (Rmax), the temperature at the maximum degradation rate (Tmax) and the residue at 700°C, are listed in Table 2.
Figure 5(a) and (b) are the TGA curves of DOPOAA and copolymers in nitrogen atmosphere from ambient temperature to 700 °C. Two decomposition stages of DOPOAA can been seen from the curves: the first stage is between 269 °C - 340 °C, and the second stage being between 347 °C - 451°C. However, in the process of neat PS degradation, the sample began to lose weight at 365 °C and stopped decomposing at 450°C, and the Rmax was -2.75 %/°C when the temperature was 414 °C, and there was nearly no residue (0.21%).
Table 2 TGA data of DOPO and copolymer in nitrogen atmosphere
|
Sample |
T5% (°C) |
Tmax1 (°C) |
Tmax2 (°C) |
Rmax1 (%/°C) |
Rmax2 (%/°C) |
Residue at 700°C (%) |
|
PS |
365 |
414 |
- |
-2.75 |
- |
0.21 |
|
DOPOAA |
269 |
307 |
414 |
-1.37 |
-0.52 |
5.62 |
|
PS-10 |
138 |
412 |
- |
-1.87 |
- |
0.89 |
|
PS-20 |
162 |
413 |
- |
-1.49 |
- |
0.91 |
(The figure 6 was uploaded in attachment.)
Figure 6. TGA(a) and DTG(b) of DOPOAA, PS and DOPO acrylate-styrene copolymer with different DOPO structure ratio in nitrogen atmosphere
At the same time, when the loading amount of DOPOAA increased, the residue at 700 °C of PS-10 and PS-20 raise to 0.89% and 0.91%. Compared with PS, the Tmax of the copolymer hardly changed, but the Rmax was reduced to -1.87 %/°C and -1.49 %/°C, reducing 32% and 45.82%, respectively. The decrease of Rmax indicates that a more compact char layer formed during the thermal degradation, and it could provide good insulation to the unburning part from the heat [21]. The change of residue at 700°C and Rmax indicated that the introduction of DOPOAA improved the thermal stability of the copolymer. In figure 4, it can be found that the T5% of PS-10 and PS-20 are lower than that of PS. This may be because the DOPO groups in copolymer catalyzed the polymer to dehydrate, and leads to more weight loss [38, 39]. However, the residue at 700°C of copolymer is higher, which indicated that copolymer has better thermal stability. Therefore, it can be considered that DOPOAA can effectively improve the thermal stability of copolymer.
Comment 5: To Furthermore, authors evaluated flame retardant behavior (in the text it is written as “…Behaviors”) by MCC and LOI test. Indeed, the results of abovementioned test are in agreement and “These MCC and LOI data clearly demonstrate that the copolymerization of DOPOAA and styrene could obviously improve the flame resistance of the PS.”. Authors are advised to carry out the vertical flame test according to DIN 53906 standard method (if applicable) and to gain more data about flame resistance.
Answer 5: Thank you very much for your suggestion. We have carefully reviewed the DIN 53906 standard method and found that it is mainly suitable for protective clothing. We will try to test the copolymer follow this standard in future studies to verify its applicability.
Thank you again for your constructive and helpful advice.

Reviewer 3 Report
The reviewed article presents an interesting and timely problem of fire properties of polymer materials, especially based on polystyrene.
The authors carried out the modification by means of copolymerization of styrene with phosphorus flame retardant. They presented the characteristics of the materials produced using the FTIR, TGA, LOI and MCC methods, among others. The discussion of the results is also poor and has a slight scientific impact. Additionally One should not conclude about the behavior towards fire from the results of TGA performed in a nitrogen atmosphere: “The char layers obstructed the substance and energy exchange between internal unburned parts and the burning parts, which can effectively prevent the spread of the flame”. These are two different mechanisms.
In my opinion, considerations about fire properties are insufficiently presented (it seems that these are preliminary studies). The article presented for review discusses the fire properties based only on the LOI test and micro-scale combustion calorimeters (MCC).
In my opinion, in order to fully characterize the fire properties, additional tests should be performed to provide complete information. The studies should be supplemented with an investigation of gaseous product emissions during the thermal decomposition, the investigation of the burning test (fire spread test) and structural investigations of the carbon residues after burning. There is also a lack of information about total smoke production (TSP) of modified PS materials.
In my opinion, given the high level of the Materials journal, I consider the results are incomplete, and they do not sufficiently describe the topic problem of the fire properties of examined materials. Therefore, I suggest that the article presented in this form should be rejected. I leave the final decision to the Editor.
Author Response
Reply to Reviewer 3
By Yazhen Wang
Thank you very much for reviewing our paper. We carefully read your comments, which are quite useful. We have made suitable revisions and explain our revisions below. We hope this revision meets the criteria to be published in Materials. Thank you again for your constructive and helpful review.
Comment 1: This manuscript reports the generation of an acrylate containing a DOPO substituent and its copolymerization with styrene in an attempt to generate flame retardant polymers. This is a worthy and timely undertaking (see Polymers, 2019, 11, 2038, doi: 10.3390/polym11122038 - this reference provides a good statement of the problem and should be cited).
Answer 1: Yes, we accepted your comments. By reading the article you mentioned conscientiously, we benefit a great deal and cited it.
Comment 2: The copolymers have not been well-characterized. What are the molecular weights, dispersities - no SEC data are provided? What is the copolymer composition? Does the acrylate go in randomly? Both questions could be answered from the quantitative C-13 NMR spectra.
Answer 2: We had measured the Mn and Mw value of copolymer through the SEC test, and the data have been added into Table 1 (line 126).
Table 1 The formulation of copolymers
|
Sample |
St (g) |
DOPOAA (g) |
DOPO content (%) |
DOPOAA in copolymer(%) |
Yield (%) |
Mn (x104) |
Mw (x104) |
|
PS |
5 |
0 |
0 |
0 |
87.6 |
3.23 |
5.69 |
|
PS-10 |
3.79 |
1.21 |
10 |
3.34 |
69 |
6.16 |
16.26 |
|
PS-20 |
2.91 |
2.01 |
20 |
5.97 |
66.3 |
6.81 |
20.97 |
From the 1H-NMR spectra of copolymer, the content of DOPOAA in the copolymer is lower than the theoretical value, this may be due to the large steric hindrance of the side chain. And the analysis of SEC and 1H-NMR of copolymer are as follows (line 140-149):
(Figure 4 has been uploaded in attachment)
Figure 4. 1H-NMR spectra of copolymer
Figure 4 is the 1H-NMR spectra of the copolymers. In the copolymer, the peaks at 3.70 correspond to the of -O-CH2-P- unit. Between 7.10 and 6.37 are the aromatic protons appeared. Signals of the main chain protons appeared at 2.21–1.22 ppm. The compositions of the copolymers were calculated from the relative areas of the methylene and main chain protons resonance, using the following formula, where B-CH2- and B0 were the relative resonance areas attributed to methylene and main chain protons. The DOPOAA content in the copolymer is shown in Table 1. It can be seen that as the increase of DOPOAA content in the feed, the DOPOAA content in the copolymer also increases. At the same time, the content of DOPOAA in the copolymer is lower than that of the feed, which may be due to the large steric hindrance of the side chain. Data from gel chromatography showed that with the introduction of DOPOAA, the molecular weight of the copolymer increased.
Comment 3: More concerning, the glass transition temperatures for the copolymers are not provided. Are the properties of poly(styrene) retained in the copolymers? At 30% DOPO loading, a very different polymer is probably formed (very different physical and mechanical properties).
Answer 3: Yes, we accepted your comments. We had tested the Tg of PS, PS-10 and PS-20 at 107℃, 103℃ and 97℃, respectively, and the data map has been added into the revised paper. See Figure 5 (line 155). In addition, we have analyzed the test data of Tg according to references No.31 and No.32 in the revised paper we upload.
(Figure 5 has been uploaded in attachment)
Figure 5. DSC curves of PS and copolymers
However, since the preparation, testing and comprehensive analysis of the sample takes too much time, our research group will test physical and mechanical properties of copolymer in future studies.
Comment 4: The manuscript will need rather massive rewriting for accuracy, completeness and readability. The Introduction needs to be expanded to better reflect the problems associated with the use of organo-halogen compounds (see attached) and the mode of action of organophosphorus flame retardants.
Corrections are penciled-in directly on pages of the manuscript attached. These are indicative of the kinds of changes needed throughout. A DOPO acrylate was not prepared "in this paper" or "in this article" but rather in the laboratory. The preparation is reported in the paper. “Free radical” should be "radical". The term "free radical" represents older nomenclature and the radicals described here are highly encumbered. "Internal standard" should be "internal reference". Careful attention should be paid to the use of articles and tenses. Author's names should be omitted.
Answer 4: Yes, we accepted your comments. We revised the manuscript conscientiously follow your comments.
Comment 5: It is suggested that thermal decomposition of DOPOAA leads to the generation of phosphoric acid and hypophosphite. There is absolutely no evidence provided to support this and it is probably incorrect. DOPOAA may eliminate a phosphorus acid but it is neither of these species. DOPO decomposes to extrude PO radical to the gas phase where it scavenges combustion propagating radicals (J. Appl. Polym. Sci., 2007,105,685-695; Chem. Eur. J., 2015, 162, 1073-1080). This doesn't occur in the solid phase.
Answer 5: Yes, we accepted your comments. We read conscientiously and cite the articles you provided as No. 40 and No.41 in reference list.
And modified this section as follows (line 198-202):
The phosphorus-containing groups in copolymer were thermally decomposed to form PO•, phosphorus-containing radical [40, 41]. The PO• radical entering the gas phase can eliminate the radicals generated in combustion, thus inhibiting the further decomposition of the copolymer. These MCC and LOI data clearly demonstrate that the copolymerization of DOPOAA and styrene could obviously improve the flame resistance of the PS.
Comment 6: The polymer structure in Scheme 1 is incorrect (there is no double bond in the mainchain).
Answer 6: Yes, you are right, please forgive my negligence, and we modified the structure of copolymer in Scheme 1 (line 87).
Scheme 1. Synthesis route of DOPO acrylate-Styrene Copolymer
Thank you again for your constructive and helpful advice.

Reviewer 4 Report
This manuscript reports the generation of an acrylate containing a DOPO substituent and its copolymerization with styrene in an attempt to generate flame retardant polymers. This is a worthy and timely undertaking (see Polymers, 2019, 11, 2038, doi: 10.3390/polym11122038 - this reference provides a good statement of the problem and should be cited). The copolymers have not been well-characterized. What are the molecular weights, dispersities - no SEC data are provided? What is the copolymer composition? Does the acrylate go in randomly? Both questions could be answered from the quantitative C-13 NMR spectra. More concerning, the glass transition temperatures for the copolymers are not provided. Are the properties of poly(styrene) retained in the copolymers? At 30% DOPO loading, a very different polymer is probably formed (very different physical and mechanical properties). It is somewhat surprising that at relatively high loading of DOPO (20,30%) a relatively modest decrease in flammability for the styrene polymers is observed (LOI, 22 and 26).
The manuscript will need rather massive rewriting for accuracy, completeness and readability. The Introduction needs to be expanded to better reflect the problems associated with the use of organohalogen compounds (see attached) and the mode of action of organophosphorus flame retardants. It is suggested that thermal decomposition of DOPOAA leads to the generation of phosphoric acid and hypophosphite. There is absolutely no evidence provided to support this and it is probably incorrect. DOPOAA may eliminate a phosphorus acid but it is neither of these species. DOPO decomposes to extrude PO radical to the gas phase where it scavenges combustion propagating radicals (J. Appl. Polym. Sci., 2007,105,685-695; Chem. Eur. J., 2015, 162, 1073-1080). This doesn't occur in the solid phase. The polymer structure in Scheme 1 is incorrect (there is no double bond in the mainchain).
Corrections are penciled-in directly on pages of the manuscript attached. These are indicative of the kinds of changes needed throughout. A DOPO acrylate was not prepared "in this paper" or "in this article" but rather in the laboratory. The preparation is reported in the paper. :Free radical" should be "radical". The term "free radical" represents older nomenclature and the radicals described here are highly encumbered. "Internal standard" should be "internal reference". Careful attention should be paid to the use of articles and tenses. Author's names should be omitted.

Author Response
Reply to Reviewer 3
By Yazhen Wang
Thank you very much for reviewing our paper. We carefully read your comments, which are quite useful. We have made suitable revisions and explain our revisions below. We hope this revision meets the criteria to be published in Materials. Thank you again for your constructive and helpful review.
Comment 1: This manuscript reports the generation of an acrylate containing a DOPO substituent and its copolymerization with styrene in an attempt to generate flame retardant polymers. This is a worthy and timely undertaking (see Polymers, 2019, 11, 2038, doi: 10.3390/polym11122038 - this reference provides a good statement of the problem and should be cited).
Answer 1: Yes, we accepted your comments. By reading the article you mentioned conscientiously, we benefit a great deal and cited it.
Comment 2: The copolymers have not been well-characterized. What are the molecular weights, dispersities - no SEC data are provided? What is the copolymer composition? Does the acrylate go in randomly? Both questions could be answered from the quantitative C-13 NMR spectra.
Answer 2: We had measured the Mn and Mw value of copolymer through the SEC test, and the data have been added into Table 1 (line 126).
Table 1 The formulation of copolymers
|
Sample |
St (g) |
DOPOAA (g) |
DOPO content (%) |
DOPOAA in copolymer(%) |
Yield (%) |
Mn (x104) |
Mw (x104) |
|
PS |
5 |
0 |
0 |
0 |
87.6 |
3.23 |
5.69 |
|
PS-10 |
3.79 |
1.21 |
10 |
3.34 |
69 |
6.16 |
16.26 |
|
PS-20 |
2.91 |
2.01 |
20 |
5.97 |
66.3 |
6.81 |
20.97 |
From the 1H-NMR spectra of copolymer, the content of DOPOAA in the copolymer is lower than the theoretical value, this may be due to the large steric hindrance of the side chain. And the analysis of SEC and 1H-NMR of copolymer are as follows (line 140-149):
(Figure 4 has been uploaded in attachment.)
Figure 4. 1H-NMR spectra of copolymer
Figure 4 is the 1H-NMR spectra of the copolymers. In the copolymer, the peaks at 3.70 correspond to the of -O-CH2-P- unit. Between 7.10 and 6.37 are the aromatic protons appeared. Signals of the main chain protons appeared at 2.21–1.22 ppm. The compositions of the copolymers were calculated from the relative areas of the methylene and main chain protons resonance, using the following formula, where B-CH2- and B0 were the relative resonance areas attributed to methylene and main chain protons. The DOPOAA content in the copolymer is shown in Table 1. It can be seen that as the increase of DOPOAA content in the feed, the DOPOAA content in the copolymer also increases. At the same time, the content of DOPOAA in the copolymer is lower than that of the feed, which may be due to the large steric hindrance of the side chain. Data from gel chromatography showed that with the introduction of DOPOAA, the molecular weight of the copolymer increased.
Comment 3: More concerning, the glass transition temperatures for the copolymers are not provided. Are the properties of poly(styrene) retained in the copolymers? At 30% DOPO loading, a very different polymer is probably formed (very different physical and mechanical properties).
Answer 3: Yes, we accepted your comments. We had tested the Tg of PS, PS-10 and PS-20 at 107℃, 103℃ and 97℃, respectively, and the data map has been added into the revised paper. See Figure 5 (line 155). In addition, we have analyzed the test data of Tg according to references No.31 and No.32 in the revised paper we upload.
(Figure 5 has been uploaded in attachment.)
Figure 5. DSC curves of PS and copolymers
However, since the preparation, testing and comprehensive analysis of the sample takes too much time, our research group will test physical and mechanical properties of copolymer in future studies.
Comment 4: The manuscript will need rather massive rewriting for accuracy, completeness and readability. The Introduction needs to be expanded to better reflect the problems associated with the use of organo-halogen compounds (see attached) and the mode of action of organophosphorus flame retardants.
Corrections are penciled-in directly on pages of the manuscript attached. These are indicative of the kinds of changes needed throughout. A DOPO acrylate was not prepared "in this paper" or "in this article" but rather in the laboratory. The preparation is reported in the paper. “Free radical” should be "radical". The term "free radical" represents older nomenclature and the radicals described here are highly encumbered. "Internal standard" should be "internal reference". Careful attention should be paid to the use of articles and tenses. Author's names should be omitted.
Answer 4: Yes, we accepted your comments. We revised the manuscript conscientiously follow your comments.
Comment 5: It is suggested that thermal decomposition of DOPOAA leads to the generation of phosphoric acid and hypophosphite. There is absolutely no evidence provided to support this and it is probably incorrect. DOPOAA may eliminate a phosphorus acid but it is neither of these species. DOPO decomposes to extrude PO radical to the gas phase where it scavenges combustion propagating radicals (J. Appl. Polym. Sci., 2007,105,685-695; Chem. Eur. J., 2015, 162, 1073-1080). This doesn't occur in the solid phase.
Answer 5: Yes, we accepted your comments. We read conscientiously and cite the articles you provided as No. 40 and No.41 in reference list.
And modified this section as follows (line 198-202):
The phosphorus-containing groups in copolymer were thermally decomposed to form PO•, phosphorus-containing radical [40, 41]. The PO• radical entering the gas phase can eliminate the radicals generated in combustion, thus inhibiting the further decomposition of the copolymer. These MCC and LOI data clearly demonstrate that the copolymerization of DOPOAA and styrene could obviously improve the flame resistance of the PS.
Comment 6: The polymer structure in Scheme 1 is incorrect (there is no double bond in the mainchain).
Answer 6: Yes, you are right, please forgive my negligence, and we modified the structure of copolymer in Scheme 1 (line 87).
Scheme 1. Synthesis route of DOPO acrylate-Styrene Copolymer
Thank you again for your constructive and helpful advice.

Round 2
Reviewer 2 Report
I am glad to confirm that the manuscript has been significantly improved and now can be published in Materials.
Author Response
We are very honored that you recognized our work. Thank you again for your constructive and helpful advice.
Reviewer 3 Report
The manuscript has been significantly improved by the authors. The entered information (DSC analysis, investigation of Mn) significantly strengthens the value of the publication. In my opinion, an implemented changes of large ranges of scientific values caused that the article should be accepted in presented form for publication in Materials
Author Response

(The authors gave the same response as above.)

Reviewer 4 Report
The manuscript is much improved. The addition of characterization data for the copolymers strengthens the paper considerably. The wording is still somewhat flawed - "were" is often used for "was", for example. Some small changes are indicated on the pages attached. The title must be corrected. These are not "polystyrene copolymers" but rather "styrene copolymers". References should be checked - the year would appear to be missing in references 31, 32.
Author Response
Reply to Reviewer 4
By Yazhen Wang
Thank you very much for reviewing our paper. We read your comments conscientiously, which are quite useful. We have made suitable revisions and explain our revisions below. We hope this revision meets the criteria to be published in Materials. Thank you again for your constructive and helpful review.
Comment 1: The wording is still somewhat flawed - "were" is often used for "was", for example.
Answer 1: Yes, we accepted your comments. The word “was” in line 20 and 22 has been replaced by “were”.
Comment 2: The title must be corrected. These are not "polystyrene copolymers" but rather "styrene copolymers".
Answer 2: Yes, you are right. The title has been modified as “The preparation, thermal properties and fire property of a phosphorus containing flame retardant styrene copolymer”.
Comment 3: References should be checked - the year would appear to be missing in references 31, 32.
Answer 3: Yes, you are right. We have checked the references conscientiously and modified 31 and 32 as follows:
Hu, W.; Zhan, J.; Hong, N.; Hull, T. R.; Stec, A. A.; Song, L.; Wang, J. Hu, Y., Flame retardant polystyrene copolymers: preparation, thermal properties, and fire toxicities. Polym. Adv. Technol. 2014, 25, 631-637. Cui, J.; Zhu, C.; He, M.; Ke, Z.; Liu, Y.; Tai, Q.; Xiao, X.; Hu, Y. Preparation of a novel styrene copolymer: Simultaneously improving the thermal stability and toughness. J. Appl. Polym. Sci., 2018, 135, 46120.
Thank you again for your constructive and helpful advice.
